# Optimal Control of Semi-Active Suspension for Agricultural Tractors Using Linear Quadratic Gaussian Control

**DOI:** 10.3390/s23146474

**Published:** 2023-07-17

**Authors:** Da-Vin Ahn, Kyeongdae Kim, Jooseon Oh, Jaho Seo, Jin Woong Lee, Young-Jun Park

**Affiliations:** 1Department of Biosystems Engineering, College of Agriculture and Life Sciences, Seoul National University, Seoul 08826, Republic of Korea; mt1125@snu.ac.kr (D.-V.A.); kkd4943@snu.ac.kr (K.K.); 2Convergence Major in Global Smart Farm, College of Agriculture and Life Sciences, Seoul National University, Seoul 08826, Republic of Korea; 3Department of Convergence Biosystems Engineering, Chonnam National University, Gwangju 61186, Republic of Korea; jooseon.oh@jnu.ac.kr; 4Department of Automotive and Mechatronics Engineering, Ontario Tech University, 2000 Simcoe Street North, Oshawa, ON L1G 0C5, Canada; jaho.seo@ontariotechu.ca; 5Smart Agricultural Machinery R&D Group, Korea Institute of Industrial Technology, Gimje 54325, Republic of Korea; reallybear@kitech.re.kr; 6Research Institute for Agriculture and Life Science, Seoul National University, Seoul 08826, Republic of Korea

**Keywords:** agricultural tractor, linear quadratic Gaussian control, ride vibration reduction, semi-active suspension, Kalman filter, observer design

## Abstract

In this study, a semi-active suspension based on a hydro-pneumatic mechanism was designed to minimize the ride vibration using a suspension control algorithm. The performance of the algorithm was critical for controlling the characteristics of the target tractor. A linear-quadratic-Gaussian (LQG) optimal control algorithm was designed as a semi-active suspension control algorithm. The plant model for developing this algorithm was based on the parameters of an actual tractor. The rear suspension deflection was represented by a Kalman-filter-based state observer feedback to estimate the state variables that were difficult to measure. The designed state observer of the LQG controller was validated in terms of an accuracy index. The estimated vertical velocity and acceleration accuracies of the cabin were 83% and 79%, respectively. The performance of the designed controller was validated in terms of a performance index by comparing the performance of a tractor equipped with a rear rubber mount with that of one equipped with a semi-active suspension. The peak and root-mean-square values of the vertical acceleration of the cabin were reduced by up to 48.97% and 47.06%, respectively. This study could serve as a basis for the application of the control algorithm to systems with similar characteristics, thereby reducing system costs.

## 1. Introduction

Tractors are used for agricultural and construction work. They are mainly used off-road where the terrain is rough; hence, low-frequency vibrations with large vibration magnitudes and exposure times are transmitted from the road surface to the tractor operator [1]. These vibrations can render the tractor operator uncomfortable and cause musculoskeletal disorders such as fatigue accumulation. A tractor is damped using tires and rubber mounts in the cabin, as it is structurally impractical to equip its wheels with a suspension system. However, reducing vibration in this manner is ineffective, in that the reduction in vibration using the rubber mounts approaches a limit as the tractor speed increases [2]. Recently, attempts have been made to reduce the effects of vibration by replacing the rubber mount with a hydro-pneumatic suspension [3].

Tractor suspension systems are classified into passive and semi-active suspension systems. In a passive suspension system, the operator arbitrarily selects the damping coefficient of the suspension. However, in a semi-active system, the optimum damping coefficient is determined using a semi-active suspension control algorithm. An effective control algorithm can significantly reduce the vibration during riding; however, the algorithm should be based on the characteristics of the target tractor.

Active and semi-active suspension systems were applied to vehicles before they were applied to tractors. Numerous studies have been conducted on the development of vehicle suspension control algorithms. Ieluzzi et al. [4] established a suspension system control strategy using MATLAB/Simulink for semi-active suspensions in trucks. The performance of this suspension was confirmed using a control algorithm developed based on semi-active suspension test data. The focus of this study was on driving stability and not on improving rider comfort, given that the suspension was mounted in the cargo compartment (not the cabin). The test was conducted under passive and semi-active suspension conditions, and a vibration reduction effect of 12% at the peak was confirmed. In addition, with a decrease in speed, the vibration in the suspension mode was found to be reduced further. Jalili [5] introduced the basic theory and concept of control logic design and the implementation of a semi-active suspension system. Fuzzy control has also been proposed [6], which is a method for mapping a discontinuous control model to a continuous linear region, as a control algorithm for a semi-active suspension control system; the design results were validated for a vehicle. Active and semi-active suspension system controls are common control technologies used in vehicles; however, applying these technologies to agricultural tractors is impractical, as the system structure of the tractor is completely different from that of a vehicle. The vehicle suspension system has a structure in which the suspension is installed between the tire (unsprung mass) and the body (sprung mass). The sprung mass is significantly larger than the unsprung mass, and the unsprung mass moves in the vertical direction [7,8,9,10,11,12]. However, in the case of a tractor, the size of its unsprung mass (tractor body) equipped with the cabin suspension is significantly larger than that of the sprung mass (tractor cabin). In addition, unlike the unsprung mass of the vehicle model, its unsprung mass moves in accordance with the vertical and pitch motions. Hence, the control algorithm of the vehicle suspension cannot be directly applied to the tractor suspension system, and the tractor suspension system is separate from the wheels. Therefore, it is necessary to implement a separate control algorithm in which the tractor characteristics are applied. Finally, for cars, the focus is on stability on the road for high-speed driving, whereas for agricultural tractors, the focus is on work stability, as they operate on various off-road agricultural sites with low-speed driving vehicles. The different targets of the suspension system control algorithm highlight the necessity for a unique tractor suspension control algorithm.

Studies on the cabin suspension of agricultural tractors initially focused on the structure and configuration of the suspension [13,14]. The focus recently shifted to the optimization of the cabin suspension and configuration of passive and semi-active control systems. Sarami [15] developed a suspension system with skyhook control applied to the front and rear of the tractor cabin. The suspension of the tractor was installed in the cabin as a semi-active suspension at the front and rear mounts. This configuration is different from that of the tractor used in this study, which adopts a semi-active suspension at the rear and rubber mount in the front. Skyhook control can achieve a high performance, even when the control input is to be calculated, except for state variables that are difficult to measure, for example, the tire displacement and road profile. However, the control, which is a full-state feedback control that requires multiple sensors, is limited in that the performance with respect to the road-holding stability is lower than that with respect to the improvement i ride comfort. A plant model, namely, a 14-degree-of-freedom (DOF) tractor model equipped with a rear semi-active suspension, has been developed [16]. The internal model uses linear quadratic Gaussian (LQG) optimal control as a quarter-car suspension model equipped with a semi-active suspension. In the aforementioned study, the quarter-car suspension model was used to design the internal model, which is critical for the development of a control algorithm with LQG control. However, it was not determined whether the behavior of the internal model used was similar to that of the plant model, that is, the 14-DOF tractor model with a semi-active suspension (in the rear mount) and rubber mount (in the front). If the behaviors of the internal model used in the suspension control algorithm and plant model of the target system are different, the system characteristics of the two models will differ. Therefore, the developed control algorithm is limited when applied to the target system. Moreover, the internal model designed for the quarter-car suspension model is limited, as implementing the behavioral characteristics of the plant model in a tractor equipped with a semi-active suspension system (in the rear mount) and rubber mount (in front) is difficult. The plant model is equipped with a semi-active suspension system in the rear mount. Moreover, the frontal rubber mount demonstrates different behavior characteristics (front suspension system) when compared with those with the rear suspension system, given that the front and rear suspension systems are not symmetric. Therefore, the pitch motion of the plant model has a significant effect on its behavior, for example, on cabin acceleration. If the internal model is implemented as a quarter-car suspension system, the effect of the pitch motion is not considered. Therefore, the internal model is limited in that it does not demonstrate the same characteristics as the plant model.

In this study, a semi-active suspension controller was designed using LQG optimal control for a tractor with non-symmetric front and rear suspension systems. This selection was made because the front suspension system employed passive suspension and the rear suspension system employed semi-active suspension. The internal model for designing the LQG control and plant model of the system was of a half-car suspension type that is free from vertical and pitch motions. Considering the actual tractor measurement environment, we developed a semi-active suspension controller using LQG optimal control in a system where full state feedback could not be achieved, as the measurable state variable was limited to a single rear suspension deflection. The LQG control used a semi-active suspension controller with a Kalman-filter-based state observer and linear quadratic regulator (LQR) control. After estimating the plant state variables that were difficult to measure using a state observer, the controller performed the LQR control algorithm using estimated state variables. The performance of the semi-active suspension controller was validated based on the accuracy of the state observer and the effect of reducing the tractor ride vibration of the LQG controller (under the ISO 8608 standard class). This study adds the following novelty to the literature:For system model design, a half-car suspension model was proposed to develop a high-fidelity plant model with a non-symmetric front and rear suspension systems (equipped with a semi-active suspension system in the rear mount and a rubber mount in front).The tractor suspension system demonstrated a behavior that was different from that of the vehicle suspension system, because the sprung mass was lighter than the unsprung mass. The tractor suspension system could help to analyze the control algorithm in the behavior of a system similar to a tractor suspension system.For the state observer design, a Kalman-filter-based state observer was formulated to estimate the state variables that were difficult or impractical to measure in a system with a limited number of measurement inputs.The semi-active suspension control algorithm (LQG optimal control) was evaluated for applicability, and critical points were highlighted for the improvement in the performance of the control algorithm.

This study makes the following contributions to the literature:A semi-active suspension-control-algorithm-based optimal control was implemented in a system in which the sprung mass was lighter than the unsprung mass, and it was used to perform a basic study to enable the application of the control algorithm to systems with similar characteristics.Although previous studies have used multiple sensors for state feedback, we designed a state observer using only one sensor to measure rear suspension deflection and to develop the control algorithm of the semi-active suspension system. This approach can reduce the system costs involved in the development of a tractor suspension control algorithm.

Overall, the empirical findings can facilitate the application of optimal control for tractor cabin suspension algorithms.

## 2. Materials and Methods

### 2.1. System Modeling

#### 2.1.1. Tractor Suspension Model

The tractor suspension model was composed of a half-car suspension model, as shown in Figure 1. The half-car suspension model was represented as a linear 4-DOF system consisting of a sprung mass (tractor cabin) connected to an unsprung mass (tractor body). The sprung and unsprung masses were free of vertical and pitch motions. The suspension system between the cabin and body consisted of front and rear suspension systems. Moreover, it was subject to road disturbance input (from the road surface) because the body was in contact with the road surface via the tire.

Two half-car suspension models were developed to compare the riding vibrations with and without the suspension control algorithm. Figure 1a presents a half-car suspension model in which the control algorithm was applied and the rear suspension was equipped with a semi-active suspension. Figure 1b depicts a half-car suspension model in which the suspension control algorithm was not applied and the rear suspension was installed with a rubber mount. The additional elements in Figure 1a,b are the same.

The front suspension system consisted of a rubber mount (passive suspension system). The rubber mount was modeled using a spring with a constant stiffness and damping coefficient. The rear suspension system of the half-car suspension model in which the control algorithm was not applied was composed of a rubber mount (passive suspension system), which was modeled using a spring of constant stiffness and damping coefficient. The rear suspension system of the half-car suspension model in which the control algorithm was applied was composed of a semi-active suspension system. The semi-active suspension system was modeled using a spring of constant stiffness and variable suspension damping coefficient. The damping coefficient of the semi-active suspension system exhibited a non-linear characteristic that varied with respect to the current of the proportional control valve. Its damping force was calculated as the product of the damping coefficient and relative velocity of the rear suspension between the cabin and body. The damping force was considered as the control input to the half-car suspension system. Finally, the front and rear tires were modeled using a spring of constant stiffness and damping coefficient. If the pitch angles of the cabin and body are sufficiently small, the vertical displacement of the cabin and body can be expressed as follows:(1)$$z_{cf}=z_{c}-L_{cf}\theta_{c}$$
(2)$$z_{cr}=z_{c}+L_{cr}\theta_{c}$$
(3)$$z_{bf}=z_{b}-L_{bf}\theta_{b}$$
(4)$$z_{br}=z_{b}+L_{br}\theta_{b}$$
(5)$$z_{tf}=z_{b}-L_{tf}\theta_{b}$$
(6)$$z_{tr}=z_{b}+L_{tr}\theta_{b}$$
where 

$z_{cf}$ = vertical displacement of the front cabin;

$z_{cr}$ = vertical displacement of the rear cabin;

$z_{bf}$ = vertical displacement of the front body;

$z_{br}$ = vertical displacement of the rear body;

$z_{tf}$ = vertical displacement of the front tire;

$z_{tr}$ = vertical displacement of the rear tire;

$z_{c}$ = vertical displacement of the center of gravity of the cabin;

$z_{b}$ = vertical displacement of the center of gravity of the body;

$L_{cf}$ = distance between the front cabin and the center of gravity of the cabin;

$L_{cr}$ = distance between the rear cabin and the center of gravity of the cabin;

$L_{bf}$ = distance between the front cabin and the center of gravity of the body;

$L_{br}$ = distance between the rear cabin and the center of gravity of the body;

$L_{tf}$ = distance between the front tire and the center of gravity of the body;

$L_{tr}$ = distance between the rear tire and the center of gravity of the body;

$\theta_{c}$ = pitch angle of the cabin;

$\theta_{b}$ = pitch angle of the body.

The equations can be expressed by applying the static equilibrium position as the vertical displacements of the centers of gravity of the cabin and body and pitch angles to a half-car suspension model (equipped with a rear semi-active suspension).

The equation of the force balance of the center of gravity of the cabin (motion for heave) can be expressed as follows:(7)$$m_{c}{\ddot{z}}_{c}=-k_{sf}\left(z_{cf}-z_{bf}\right)-c_{sf}\left({\dot{z}}_{cf}-{\dot{z}}_{bf}\right)-k_{sr}\left(z_{cr}-z_{br}\right)+f_{s}$$

The moment equation of balance of the center of gravity of the cabin (motion for pitch) can be expressed as follows:(8)$$I_{c}{\ddot{\theta }}_{c}=k_{sf}\left(z_{cf}-z_{bf}\right)L_{cf}-k_{sr}\left(z_{cr}-z_{br}\right)L_{cr}+c_{sf}\left({\dot{z}}_{cf}-{\dot{z}}_{bf}\right)L_{cf}+f_{s}L_{cr}$$
where 

$m_{c}$ = mass of the tractor cabin;

$I_{c}$ = centroidal moment of inertia of the tractor cabin;

$k_{sf}$ = spring stiffness of the front rubber mount;

$c_{sf}\;$= damping coefficient of the front rubber mount;

$k_{sr}$ = spring stiffness of the rear semi-active suspension;

$f_{s}$ = suspension force of the rear semi-active suspension.

The force balance equation of the center of gravity of the body (motion for heave) can be expressed as follows:(9)$$m_{b}{\ddot{z}}_{b}=k_{sf}\left(z_{cf}-z_{bf}\right)+k_{sr}\left(z_{cr}-z_{br}\right)+c_{sf}\left({\dot{z}}_{cf}-{\dot{z}}_{bf}\right)-f_{s}-k_{tf}\left(z_{tf}-z_{rf}\right)-k_{tr}\left(z_{tr}-z_{rr}\right)\;-c_{tf}\left({\dot{z}}_{tf}-{\dot{z}}_{rf}\right)-c_{tr}\left({\dot{z}}_{tr}-{\dot{z}}_{rr}\right).$$

The equation of moment balance at the center of gravity of the body (motion for pitch) can be expressed as follows:(10)$$I_{b}{\ddot{\theta }}_{b}=k_{tf}\left(z_{tf}-z_{rf}\right)L_{tf}-k_{tr}\left(z_{tr}-z_{rr}\right)L_{tr}+c_{tf}\left({\dot{z}}_{tf}-{\dot{z}}_{rf}\right)L_{tf}-c_{tr}\left({\dot{z}}_{tr}-{\dot{z}}_{rr}\right)L_{tr}-k_{sf}\left(z_{cf}-z_{bf}\right)L_{bf}\;+k_{sr}\left(z_{cr}-z_{br}\right)L_{br}-c_{sf}\left({\dot{z}}_{cf}-{\dot{z}}_{bf}\right)L_{bf}-f_{s}L_{br}.$$
where 

$m_{b}$ = mass of the tractor body;

$I_{b}$ = centroidal moment of inertia in the tractor body;

$z_{rf}$ = front road vertical profile;

$z_{rr}$ = rear road vertical profile;

$k_{tf}$ = spring stiffness of the front tire;

$c_{tf}$ = damping coefficient of the front tire;

$k_{tr}$ = spring stiffness of the rear tire;

$c_{tr}$ = damping coefficient of the rear tire.

Using Equations (7)–(10), the dynamic equation can be expressed in matrix form as follows:(11)$$\left[\begin{array}{cc} m_{c} & 0\\ 0 & I_{c} \end{array}\right]\left[\begin{array}{c} {\ddot{z}}_{c}\\ {\ddot{\theta }}_{c} \end{array}\right]=-\left[\begin{array}{cc} 1 & 1\\ -L_{cf} & L_{cr} \end{array}\right]\left[\begin{array}{cc} k_{sf} & 0\\ 0 & k_{sr} \end{array}\right]\left[\begin{array}{c} z_{cf}-z_{bf}\\ z_{cr}-z_{br} \end{array}\right]-\left[\begin{array}{c} \begin{array}{cc} 1 & 1\\ -L_{cf} & L_{cr} \end{array} \end{array}\right]\left[\begin{array}{cc} c_{sf} & 0\\ 0 & 0 \end{array}\right]\left[\begin{array}{c} {\dot{z}}_{cf}-{\dot{z}}_{bf}\\ {\dot{z}}_{cr}-{\dot{z}}_{br} \end{array}\right]+\left[\begin{array}{cc} 1 & 1\\ -L_{cf} & L_{cr} \end{array}\right]\left[\begin{array}{c} 0\\ f_{s} \end{array}\right]$$
(12)$$\left[\begin{array}{cc} m_{b} & 0\\ 0 & I_{b} \end{array}\right]\left[\begin{array}{c} {\ddot{z}}_{b}\\ {\ddot{\theta }}_{b} \end{array}\right]=\left[\begin{array}{cc} 1 & 1\\ -L_{bf} & L_{br} \end{array}\right]\left[\begin{array}{cc} k_{sf} & 0\\ 0 & k_{sr} \end{array}\right]\left[\begin{array}{c} z_{cf}-z_{bf}\\ z_{cr}-z_{br} \end{array}\right]+\left[\begin{array}{c} \begin{array}{cc} 1 & 1\\ -L_{bf} & L_{br} \end{array} \end{array}\right]\left[\begin{array}{cc} c_{sf} & 0\\ 0 & 0 \end{array}\right]\left[\begin{array}{c} {\dot{z}}_{cf}-{\dot{z}}_{bf}\\ {\dot{z}}_{cr}-{\dot{z}}_{br} \end{array}\right]\;-\left[\begin{array}{cc} 1 & 1\\ -L_{tf} & L_{tr} \end{array}\right]\left[\begin{array}{cc} k_{tf} & 0\\ 0 & k_{tr} \end{array}\right]\left[\begin{array}{c} z_{tf}-z_{rf}\\ z_{tr}-z_{rr} \end{array}\right]-\left[\begin{array}{c} \begin{array}{cc} 1 & 1\\ -L_{tf} & L_{tr} \end{array} \end{array}\right]\left[\begin{array}{cc} c_{tf} & 0\\ 0 & c_{tr} \end{array}\right]\left[\begin{array}{c} {\dot{z}}_{tf}-{\dot{z}}_{rf}\\ {\dot{z}}_{tr}-{\dot{z}}_{rr} \end{array}\right]-\left[\begin{array}{cc} 1 & 1\\ -L_{bf} & L_{br} \end{array}\right]\left[\begin{array}{c} 0\\ f_{s} \end{array}\right].$$

The aforementioned matrix can be expressed as follows:(13)$$M_{c}{\ddot{X}}_{c}=-R_{c}K_{s}\left(Z_{c}-Z_{b}\right)-R_{c}C_{sf}\left({\dot{Z}}_{c}-{\dot{Z}}_{b}\right)+R_{c}F_{s}$$
(14)$$M_{b}{\ddot{X}}_{b}=R_{b}K_{S}\left(Z_{c}-Z_{b}\right)+R_{b}C_{sf}\left({\dot{Z}}_{c}-{\dot{Z}}_{b}\right)-R_{t}K_{t}\left(Z_{t}-Z_{r}\right)-R_{t}C_{t}\left({\dot{Z}}_{t}-{\dot{Z}}_{r}\right)-R_{b}F_{s}$$
where

$M_{c}=\left[\begin{array}{cc} m_{c} & 0\\ 0 & I_{c} \end{array}\right]$, $M_{b}=\left[\begin{array}{cc} m_{b} & 0\\ 0 & I_{b} \end{array}\right]$, ${\ddot{X}}_{c}=\left[\begin{array}{c} {\ddot{z}}_{c}\\ {\ddot{\theta }}_{c} \end{array}\right]$, ${\ddot{X}}_{b}=\left[\begin{array}{c} {\ddot{z}}_{b}\\ {\ddot{\theta }}_{b} \end{array}\right]$, $R_{c}=\left[\begin{array}{cc} 1 & 1\\ -L_{cf} & L_{cr} \end{array}\right]$, $R_{b}=\left[\begin{array}{cc} 1 & 1\\ -L_{bf} & L_{br} \end{array}\right]$, $R_{t}=\left[\begin{array}{cc} 1 & 1\\ -L_{tf} & L_{tr} \end{array}\right]$, $K_{s}=\left[\begin{array}{cc} k_{sf} & 0\\ 0 & k_{sr} \end{array}\right]$, $C_{sf}=\left[\begin{array}{cc} c_{sf} & 0\\ 0 & 0 \end{array}\right]$, $K_{t}=\left[\begin{array}{cc} k_{tf} & 0\\ 0 & k_{tr} \end{array}\right]$, $C_{t}=\left[\begin{array}{cc} c_{tf} & 0\\ 0 & c_{tr} \end{array}\right]$, $Z_{c}=\left[\begin{array}{c} z_{cf}\\ z_{cr} \end{array}\right],$ $Z_{b}=\left[\begin{array}{c} z_{bf}\\ z_{br} \end{array}\right]$, $Z_{t}=\left[\begin{array}{c} z_{tf}\\ z_{tr} \end{array}\right],\;Z_{r}=\left[\begin{array}{c} z_{rf}\\ z_{rr} \end{array}\right],\;{\dot{Z}}_{c}=\left[\begin{array}{c} {\dot{z}}_{cf}\\ {\dot{z}}_{cr} \end{array}\right],\;{\dot{Z}}_{b}=\left[\begin{array}{c} {\dot{z}}_{bf}\\ {\dot{z}}_{br} \end{array}\right],\;{\dot{Z}}_{r}=\left[\begin{array}{c} {\dot{z}}_{tf}\\ {\dot{z}}_{tr} \end{array}\right],\;{\dot{Z}}_{r}=\left[\begin{array}{c} {\dot{z}}_{rf}\\ {\dot{z}}_{rr} \end{array}\right]$, and $F_{s}=\left[\begin{array}{c} 0\\ f_{s} \end{array}\right]$.

The variable of the center, upon multiplication with the transform matrix, is transformed into the variables of the front and rear suspensions as follows:(15)$$Z_{c}=R_{c}^{T}X_{c}$$
(16)$$Z_{b}=R_{b}^{T}X_{b}$$
(17)$$Z_{t}=R_{t}^{T}X_{b}$$

Using the transformation matrix and Equations (15)–(17), the dynamic equation and variables can be expressed as follows:(18)$${\ddot{X}}_{c}={\left(R_{c}^{T}\right)}^{-1}{\ddot{Z}}_{c}$$
(19)$${\ddot{X}}_{b}={\left(R_{t}^{T}\right)}^{-1}{\ddot{Z}}_{t}$$
(20)$${\dot{Z}}_{b}=R_{b}^{T}{\left(R_{t}^{T}\right)}^{-1}{\dot{Z}}_{t}$$
where 

$X_{c}=\left[\begin{array}{c} z_{c}\\ \theta_{c} \end{array}\right],\;X_{b}=\left[\begin{array}{c} z_{b}\\ \theta_{b} \end{array}\right],\;{\ddot{Z}}_{c}=\left[\begin{array}{c} {\ddot{z}}_{cf}\\ {\ddot{z}}_{cr} \end{array}\right]$, and ${\ddot{Z}}_{t}=\left[\begin{array}{c} {\ddot{z}}_{tf}\\ {\ddot{z}}_{tr} \end{array}\right]$.
(21)$${\ddot{Z}}_{c}=-R_{c}^{T}{\left(M_{c}\right)}^{-1}R_{c}\left[K_{s}\left(Z_{c}-Z_{b}\right)+C_{sf}{\dot{Z}}_{c}-C_{sf}R_{b}^{T}{\left(R_{t}^{T}\right)}^{-1}{\dot{Z}}_{t}\right]+R_{c}^{T}{\left(M_{c}\right)}^{-1}R_{c}F_{s}$$(22)$${\ddot{Z}}_{t}=R_{t}^{T}{\left(M_{b}\right)}^{-1}R_{b}\left[K_{s}\left(Z_{c}-Z_{b}\right)+C_{sf}{\dot{Z}}_{c}-C_{sf}R_{b}^{T}{\left(R_{t}^{T}\right)}^{-1}{\dot{Z}}_{t}\right]-R_{t}^{T}{\left(M_{b}\right)}^{-1}R_{t}\left[K_{t}\left(Z_{t}-Z_{r}\right)+C_{t}{\dot{Z}}_{t}\right]\;+R_{t}^{T}{\left(M_{b}\right)}^{-1}R_{t}C_{t}{\dot{Z}}_{r}-R_{t}^{T}{\left(M_{b}\right)}^{-1}R_{b}F_{s}$$

The state variables of the half-car suspension model are defined as follows: $x_{1}=z_{cf}-z_{bf}$ is the front suspension deflection; $x_{2}=z_{cr}-z_{br}$ is the rear suspension deflection; $x_{3}={\dot{z}}_{cf}$ is the front cabin velocity; $x_{4}={\dot{z}}_{cr}$ is the rear cabin velocity; $x_{5}=z_{tf}-z_{rf}$ is the front tire deflection; $x_{6}=z_{tr}-z_{rr}$ is the rear tire deflection; $x_{7}={\dot{z}}_{tf}$ is the front tire velocity; and $x_{8}={\dot{z}}_{tr}$ is the rear tire velocity. The state equation with the state variables of the half-car suspension model can be expressed as follows:(23)$$\left[\begin{array}{c} {\dot{x}}_{1}\\ {\dot{x}}_{2} \end{array}\right]=\left[\begin{array}{c} x_{3}\\ x_{4} \end{array}\right]-R_{b}^{T}{\left(R_{t}^{T}\right)}^{-1}\left[\begin{array}{c} x_{7}\\ x_{8} \end{array}\right]$$
(24)$$\left[\begin{array}{c} {\dot{x}}_{3}\\ {\dot{x}}_{4} \end{array}\right]=-R_{c}^{T}{\left(M_{c}\right)}^{-1}R_{c}K_{r}\left[\begin{array}{c} x_{1}\\ x_{2} \end{array}\right]-R_{c}^{T}{\left(M_{c}\right)}^{-1}R_{c}C_{sf}\left[\begin{array}{c} x_{3}\\ x_{4} \end{array}\right]+R_{c}^{T}{\left(M_{c}\right)}^{-1}R_{c}C_{sf}R_{b}^{T}{\left(R_{t}^{T}\right)}^{-1}\left[\begin{array}{c} x_{7}\\ x_{8} \end{array}\right]+R_{c}^{T}{\left(M_{c}\right)}^{-1}R_{c}F_{s}$$
(25)$$\left[\begin{array}{c} {\dot{x}}_{5}\\ {\dot{x}}_{6} \end{array}\right]=\left[\begin{array}{c} x_{7}\\ x_{8} \end{array}\right]-{\dot{Z}}_{r}$$
(26)$$\left[\begin{array}{c} {\dot{x}}_{7}\\ {\dot{x}}_{8} \end{array}\right]=R_{t}^{T}{\left(M_{b}\right)}^{-1}R_{b}K_{s}\left[\begin{array}{c} x_{1}\\ x_{2} \end{array}\right]+R_{t}^{T}{\left(M_{b}\right)}^{-1}R_{b}C_{sf}\left[\begin{array}{c} x_{3}\\ x_{4} \end{array}\right]-R_{t}^{T}{\left(M_{b}\right)}^{-1}R_{t}K_{t}\left[\begin{array}{c} x_{5}\\ x_{6} \end{array}\right]\;+\left[R_{t}^{T}{\left(M_{b}\right)}^{-1}R_{t}C_{t}-R_{t}^{T}{\left(M_{b}\right)}^{-1}R_{b}C_{sf}R_{b}^{T}{\left(R_{t}^{T}\right)}^{-1}\right]\left[\begin{array}{c} x_{7}\\ x_{8} \end{array}\right]-R_{t}^{T}{\left(M_{b}\right)}^{-1}R_{b}F_{s}+R_{t}^{T}{\left(M_{b}\right)}^{-1}R_{t}C_{t}{\dot{Z}}_{r}$$

The aforementioned equation can be expressed as the following state-space equation:(27)$$\dot{x}=Ax+BF_{s}+L{\dot{Z}}_{r}$$
where
$$A=\left[\begin{array}{cccc} 0 & I & 0 & -R_{b}^{T}{\left(R_{t}^{T}\right)}^{-1}\\ -R_{c}^{T}{\left(M_{c}\right)}^{-1}R_{c}K_{s} & -R_{c}^{T}{\left(M_{c}\right)}^{-1}R_{c}C_{sf} & 0 & R_{c}^{T}{\left(M_{c}\right)}^{-1}R_{c}C_{sf}R_{b}^{T}{\left(R_{t}^{T}\;\right)}^{-1}\\ 0 & 0 & 0 & I\\ R_{t}^{T}{\left(M_{b}\right)}^{-1}R_{b}K_{s} & R_{t}^{T}{\left(M_{b}\right)}^{-1}R_{b}C_{sf} & -R_{t}^{T}{\left(M_{b}\right)}^{-1}R_{t}K_{t} & -R_{t}^{T}{\left(M_{b}\right)}^{-1}R_{b}C_{sf}R_{b}^{T}{\left(R_{t}^{T}\right)}^{-1}+R_{t}^{T}{\left(M_{b}\right)}^{-1}R_{t}C_{t} \end{array}\right]$$
$$B=\left[\begin{array}{c} 0\\ R_{c}^{T}{\left(M_{C}\right)}^{-1}R_{c}\\ 0\\ R_{t}^{T}{\left(M_{b}\right)}^{-1}R_{b} \end{array}\right],\;\mathrm{and}\;L=\left[\begin{array}{c} 0\\ 0\\ -I\\ R_{t}^{T}{\left(M_{b}\right)}^{-1}R_{t}C_{t} \end{array}\right]$$

The state-space equation with state variables for a half-car suspension model with a rear rubber mount is almost the same as that of a half-car suspension model with a rear semi-active suspension. The difference between the state-space equation of a half-car suspension model with a rear semi-active suspension and one with a rear rubber mount is dependent on whether the damping force is considered as a control input to the system. The damping force of the rear suspension, which is considered as the control input in a half-car suspension model equipped with a rear semi-active suspension, is not considered as the control input to one equipped with a rear rubber mount. Therefore, the state-space equation of the half-car suspension model equipped with a rear rubber mount can be expressed as follows:(28)$$\dot{x}=A_{rub}x+L{\dot{Z}}_{r}$$
where,
$$A_{rub}=\left[\begin{array}{cccc} 0 & I & 0 & -R_{sl}^{T}{\left(R_{t}^{T}\right)}^{-1}\\ -R_{c}^{T}{\left(M_{c}\right)}^{-1}R_{c}K_{rub} & -R_{c}^{T}{\left(M_{c}\right)}^{-1}R_{c}C_{rub} & 0 & R_{c}^{T}{\left(M_{c}\right)}^{-1}R_{c}C_{rub}R_{b}^{T}{\left(R_{t}^{T}\;\right)}^{-1}\\ 0 & 0 & 0 & I\\ R_{t}^{T}{\left(M_{b}\right)}^{-1}R_{b}K_{rub} & R_{t}^{T}{\left(M_{b}\right)}^{-1}R_{b}C_{rub} & -R_{t}^{T}{\left(M_{b}\right)}^{-1}R_{t}K_{t} & -R_{t}^{T}{\left(M_{b}\right)}^{-1}R_{b}C_{rub}R_{b}^{T}{\left(R_{t}^{T}\right)}^{-1}+R_{t}^{T}{\left(M_{b}\right)}^{-1}R_{t}C_{t} \end{array}\right]$$
$$K_{rub}=\left[\begin{array}{cc} k_{sf} & 0\\ 0 & k_{rr} \end{array}\right],\;and\;C_{rub}=\left[\begin{array}{cc} c_{sf} & 0\\ 0 & c_{rr} \end{array}\right]$$

$k_{rr}$ = spring stiffness of the rear rubber mount;

$c_{rr}$ = damping coefficient of the rear rubber mount.

#### 2.1.2. Semi-Active Suspension System 

The semi-active suspension system demonstrated two characteristics, as follows: 

(1) The suspension functioned only in the energy-dissipating direction of the system.

(2) The magnitude and direction of the damping force were determined using the non-linear characteristic curve with respect to the current of the proportional control valve.

Characteristic (1)

The mechanical power (P) of the semi-active suspension system is defined by Equation (29) using the control input ($f_{c}$) and relative velocity (${\dot{z}}_{cr}-{\dot{z}}_{br}$) of the suspension determined by the control law, where P≥0 indicates that the energy of the system is dissipated, and P<0 indicates that the energy is supplied to the system. An active suspension can apply a control input $f_{c}$ to the system regardless of the sign of P; however, a semi-active suspension can apply a control input $f_{c}$ only when P≥0 [17].
(29)$$P=-f_{c}\left({\dot{z}}_{cr}-{\dot{z}}_{br}\right)$$

Characteristic (2)

The magnitude and direction of the damping force are determined using the damping coefficient $c_{sr}$ and the relative velocity of the rear suspension $\left({\dot{z}}_{cr}-{\dot{z}}_{br}\right)$ from Equation (30). The suspension deflection is measured using a linear variable differential transformer (LVDT) in a semi-active suspension system. The relative velocity of the rear suspension is calculated by differentiating the measured rear suspension deflection. The damping coefficient was determined by the opening position of the orifice in the proportional control valve, which in turn was controlled by the proportional control valve current. Therefore, the damping force changed nonlinearly with respect to the valve current and the relative velocity of the rear suspension [17].
(30)$$f_{s}=-c_{sr}\left({\dot{z}}_{cr}-{\dot{z}}_{br}\right)$$
where

$c_{sr}$ = damping coefficient of the rear semi-active suspension.

The suspension damping force $f_{s}$ is determined using Characteristics (1) and (2), as expressed by Equation (31). When P≥0, $f_{s}$ is determined by the control input $f_{c}$. When P<0, $f_{s}$ is not controllable; therefore, it is determined not by $f_{c}$, but by the damping coefficient $c_{pre}$ from the previous step. The magnitude of $f_{c}$ is determined only by $c_{sr}$ because $f_{c}$ is a product of $c_{sr}$ and the relative velocity of the rear suspension. Therefore, $f_{s}$ is controlled using the damping coefficient $c_{sr}$ and damping coefficient $c_{pre}$ from the previous step.
(31)$$f_{s}=\left\{\begin{array}{cc} -c_{sr}\left({\dot{z}}_{cr}-{\dot{z}}_{br}\right)=f_{c} & if\;P\geq 0\\ -c_{pre}\left({\dot{z}}_{cr}-{\dot{z}}_{br}\right) & if\;P<0 \end{array}\right.$$

The semi-active suspension used in this study demonstrated the stiffness and damping characteristics shown in Figure 2, and the characteristics of the semi-active suspension derived from a study by [18] were adopted.

#### 2.1.3. Road Excitation

A standard road profile from ISO-8608 was adopted as the external road excitation for the simulation model [19]. In the ISO-8608 standard, random road surface profiles are classified under Classes A–H. The classification is based on the power spectral density (PSD) of the road profile, which can be expressed as follows:(32)$$G_{d}\left(n\right)=G_{d}\left(n_{0}\right){\left(\frac{n}{n_{0}}\right)}^{-\omega }$$
where 

n (cycles/m) = spatial frequency

$n_{0}$ (cycles/m) = reference spatial frequency

$G_{d}$ ($m^{3}$) = road displacement PSD

ω = exponent of the fitted PSD (for most of the road surface ω=2). 

Table 1 lists the PSDs for different road classes.

In this simulation, the road profiles from Classes A–C were used for the road excitation of the front tire. For the rear tire, the same profile class as that of the corresponding front tire was used with a certain time delay Δt, which can be expressed as follows:(33)$$\Delta t=L_{wb}/v$$
where

$L_{wb}$ (m) = length of the wheelbase;

v (m/s) = speed of the tractor.

The tractor speed is 5 km/h, which is the standard speed for rotary work, and the road profile used in the simulation is shown in Figure 3. 

### 2.2. LQG Control

The objective of the control algorithm was to improve ride comfort; the vertical acceleration of the vehicle body was quantified as the target of the control algorithm [20]. In the half-car suspension system tractor model, the vertical acceleration of the cabin was quantified as a target of the control algorithm, and the acceleration of the cabin was expressed as a state variable, as follows:(34)$${\ddot{z}}_{c}=C_{acc}x+D_{acc}u$$
(35)$$C_{acc}=\left[\begin{array}{cccc} -M_{c}^{-1}R_{c}K_{s} & -M_{c}^{-1}R_{c}C_{sf} & 0 & M_{c}^{-1}R_{c}C_{sf}R_{b}^{T}{\left(R_{t}^{T}\right)}^{-1} \end{array}\right]$$
(36)$$D_{acc}=M_{c}^{-1}R_{c}$$

The control algorithm in this study was designed as a Kalman filter-based state observer with optimal control using LQR to improve the ride comfort of the tractor half-car suspension system. The LQR controller was a full-state feedback control logic that required all of the state variables. Therefore, in this study, to use the LQR controller in a system that could measure only limited state variables, it was necessary to estimate other state variables that were difficult to measure. These variables were estimated using the state observer to feed back the limited number of state variables. If the estimated state variables exhibited an acceptable level of accuracy, they could be used as feedback state variables in the LQR controller. The Kalman-filter-based state observer enabled the estimation of state variables that could not be directly measured in a system with process and measurement noises. Thus, the result of applying the full state variables estimated from the Kalman-filter-based state observer to the LQR controller, which demonstrated full-state feedback control, was referred to as optimal sensor-based feedback. A controller that combined LQR control with full-state feedback using Kalman filter resulting in LQG control was used by [21]. In this study, a semi-active suspension control using an LQG controller was performed, and the block diagram of the LQG controller is shown in Figure 4. 

#### 2.2.1. State Observer Design

A state observer enabled the estimation of state variables with an acceptable level of accuracy for state variables that were difficult or impractical to measure in a system wherein only a limited number of sensors were installed. In this study, the suspension deflection and suspension velocity of the rear suspension of the tractor were measured and used as the feedback state variables for the Kalman-filter-based state observer. The state variables that were difficult or impractical to measure and the vertical acceleration of the cabin as the control target were estimated by the state observer. The deflection of the rear suspension was measured using the LVDT at the rear suspension system, and the suspension velocity of the rear suspension was derived by differentiating the measured suspension deflection. The system model can be expressed as a state-space equation as follows:(37)$$\dot{x}=Ax+Bu+L{\dot{z}}_{r}$$
(38)y=Cx+Du+v
where 

${\dot{z}}_{r}$ = road profile (road velocity) and process noise.

v = sensor measurement noise.



$C=\left[\begin{array}{cccccccc} 0 & 1 & 0 & 0 & 0 & 0 & 0 & 0 \end{array}\right]$





$D=\left[0\right]$



The estimated state variable $\hat{x}$ can be calculated using the estimator dynamical system and expressed as follows: (39)$$\frac{d}{dt}\hat{x}=A\hat{x}+Bu+K_{f}\left(y-\hat{y}\right)$$
(40)$$\hat{y}=C\hat{x}+Du$$
where 

$K_{f}$ = Kalman filter gain.

The Kalman filter gain was determined as the value at which the cost function was minimized [21] and can be expressed as follows:(41)$$J=\underset{t\to \infty }{\lim}E\left[{\left(x_{k}-{\hat{x}}_{k}\right)}^{2}\right]$$
(42)$$K_{f}=YC^{T}V_{n}$$
where 

E = expected value.

Y = solution of the following algebraic Riccati equation:(43)$$YA^{T}+AY-YC^{T}V_{n}^{-1}CY+V_{d}=0$$
where 

$V_{d}$ and $V_{n}$ are positive semi-definite terms with entries containing the covariances of the process and measurement noise terms.

#### 2.2.2. Optimal Control Based on a State Observer

After estimating the plant state variables using a state observer, the controller calculated the LQR control algorithm using the estimated state variables. The LQR controller determined the control input (rear suspension force $f_{c}$) that minimized the cost function, and it was expressed as follows:(44)$$J_{f_{c}}=\int_{0}^{\infty }({\hat{x}}^{T}Q\hat{x}+2{\hat{x}}^{T}NF_{c}+F_{c}^{T}RF_{c})dt$$

In this study, the cost function was defined as the state weight of the cost function, including the cabin vertical acceleration, cabin vertical velocity, state variable, and control input:$$J_{f_{c}}=\int_{0}^{\infty }({\hat{x}}^{T}Q_{0}\hat{x}+{\hat{x}}^{T}Q_{{\dot{z}}_{c}}\hat{x}+{\hat{x}}^{T}Q_{{\ddot{z}}_{c}}\hat{x}+2{\hat{x}}^{T}NF_{c}+F_{c}^{T}RF_{c})dt$$
(45)$$=\int_{0}^{\infty }({\hat{x}}^{T}Q\hat{x}+2{\hat{x}}^{T}NF_{c}+F_{c}^{T}RF_{c})dt$$
(46)$$Q=Q_{0}+Q_{{\dot{z}}_{c}}+Q_{{\ddot{z}}_{c}}$$
where 

$Q_{0}$ = state weight matrix of the state variables; 

$Q_{{\dot{z}}_{c}}$ = state weight matrix of the cabin vertical velocity; 

$Q_{{\ddot{z}}_{c}}$ = state weight matrix of the cabin vertical acceleration. 

The state-weight matrices for this system are presented in Appendix A.

The control input $f_{c}$, which was determined to minimize the cost function using Equation (45), was calculated using G, derived using the Riccati equation, and the estimated state variable $\hat{x},$ which was derived by the Kalman-filter-based state observer [17]:(47)$$F_{c}=-G\hat{x}$$
(48)$$G=R^{-1}\left(B^{T}P+N\right)$$
(49)$${\left(A-BR^{-1}N\right)}^{T}P+P\left(A-BR^{-1}N\right)+\left(Q-S^{T}R^{-1}S\right)-PBR^{-1}B^{T}P=0$$

Figure 5 presents a flowchart of the LQG controller for the semi-active suspension system of the tractor. The steps for implementing the framework of the LQG controller (for the tractor semi-active suspension system) were as follows.

(1) When the road profile, that is, process noise (disturbance), and the semi-active suspension force were applied to the half-car suspension system of the tractor, the rear suspension deflection was measured by adding sensor noise (v) to the LVDT sensor of the rear suspension system. The suspension velocity of the rear suspension was calculated by removing the sensor noise using a low-pass filter to measure the rear suspension deflection due to the addition of sensor noise.

(2) The Kalman-filter-based state observer estimated the state variables that were difficult or impractical to measure using the suspension deflection and velocity of the rear suspension as feedback state variables.

(3) The state variables estimated by the state observer were used as the state variables of the LQR control algorithm. The LQR control algorithm calculated the target force of the rear suspension ($f_{c}$), for which the cost function expressed in Equation (45) was minimized.

(4) The rear suspension force ($f_{s}$) was calculated using Equation (31) based on Characteristics (1) and (2) for the semi-active suspension.

The calculated rear suspension force was added as the control input to the half-car suspension system of the tractor with process noise (disturbance), that is, the road profile. Subsequently, the process was initiated at Step 1, in which the deflection and velocity of the rear suspension were measured as outputs using the LVDT sensor in the rear suspension system.

## 3. Results

The semi-active suspension control system was validated by conducting a MATLAB/Simulink simulation. The accuracy and performance indices were defined for the quantitative performance verification of the state observer and LQG control algorithm. The tractor parameters used in the simulation are listed in Table 2.

### 3.1. State Observer Validation

The measurement input of the half-car suspension system used in this study was only one rear suspension deflection measured in the LVDT of the rear suspension system, and the state observer was designed using only this measurement. For the quantitative analysis of the Kalman-filter-based state observer, the accuracy index can be defined as follows [22]:(50)$$Accuracy=\left[1-\frac{\sqrt{\sum_{k=1}^{N}{\left(x\left(k\right)-\hat{x}\left(k\right)\right)}^{2}}}{\sqrt{\sum_{k=1}^{N}{\left(x\left(k\right)\right)}^{2}}}\right]\times 100\%$$
where x is the system state information, $\hat{x}$ is the corresponding estimated value, and *N* is the number of samples.

With an increase in the accuracy index, the estimated state variable approaches the system state variable, which indicates a high accuracy of the state observer. For the road profile for state observer validation, a Class A ISO 8608 standard road profile was adopted as the process noise (road profile), and the speed of the tractor was maintained at 5 km/h. The state observer validation compared the system state (x), response of the tractor suspension system, estimated state ($\hat{x}$), and response of the state observer as the road profile was applied to the tractor suspension system. A schematic of the state observer validation is shown in Figure 6.

The system states for the state observer validation were the rear suspension deflection, relative velocity of the rear suspension, cabin vertical velocity, and cabin vertical acceleration. The time-domain accuracy results of the system states are shown in Figure 7, Figure 8, Figure 9 and Figure 10, and the accuracy indices of the state observer are listed in Table 3, where the system states are the rear suspension deflection, relative velocity of the rear suspension, cabin vertical velocity, and cabin vertical acceleration for a Class A ISO 8608 standard road profile with a tractor speed of 5 km/h. The accuracy results under the Class B and C ISO 8608 standard road profiles are presented in Appendix B.

The accuracies of the deflection and relative velocity of the rear suspension were higher than those of the other system states for state observer validation, given that the rear suspension deflection was a system state measured using a sensor mounted as a measurement input of the system, and the relative velocity of the rear suspension was a system state calculated by differentiating the measured rear suspension deflection. The accuracies of the cabin vertical velocity and the cabin vertical acceleration were 83.31% and 79.87%, respectively, which were lower than those of the deflection and relative velocity of the rear suspension. These differences can be attributed to the limitations in estimating the state variables used to calculate the vertical velocity and acceleration of the cabin in the half-car suspension system state-space model. The results reveal that a relatively lower accuracy was achieved because the number of measurement inputs were too few to estimate the state variables used for the state observer (by measuring only the rear suspension deflection) and measurement input of the half-car suspension system. However, the estimated state variables of the state observer (constructed using a limited number of measurement inputs) yielded different system state values. In particular, they were estimated without a time delay, as shown in Figure 9 and Figure 10. Therefore, using a state observer for the state feedback of the LQR control was acceptable.

### 3.2. LQG Controller Simulation Results

To demonstrate the performance of the LQR controller, we defined the performance indices as follows [23]:

(1) Peak value of the cabin acceleration.

(2) Root mean square (RMS) value of the cabin acceleration:(51)$$RMS\left({\ddot{z}}_{c}\right)=\sqrt{\frac{1}{k}\sum_{i=1}^{k}{\left({\ddot{z}}_{c}\left(i\right)\right)}^{2}}$$

The performance indices were evaluated by comparing the responses of tractors with a rear rubber mount, passive suspension system, and rear semi-active suspension system for cabin acceleration. The ISO standard 8608 Classes A–C were adopted for the excitation road profile, and the speed of the tractor was maintained at 5 km/h.

The time-domain performance index results for the vertical acceleration of the cabin between the semi-active suspension and rubber mount under ISO 8608 standard Classes A–C with a tractor speed of 5 km/h are shown in Figure 11, Figure 12 and Figure 13, and the performance index results for the ISO 8608 standard road profile are listed in Table 4, Table 5 and Table 6. For a quantitative analysis of the performance index results, the reduction ratio of the cabin acceleration was calculated as follows:(52)$$Reduction\;ratio=\frac{{\ddot{z}}_{c_{passive}}-{\ddot{z}}_{c_{semi-active}}}{{\ddot{z}}_{c_{passive}}}\times 100\;\left(\%\right)$$
where

${\ddot{z}}_{c_{passive}}$ = vertical acceleration of the passive suspension system of the cabin;

${\ddot{z}}_{c_{semi-active}}$ = vertical acceleration of the semi-active suspension system of the cabin.

**Figure 11 sensors-23-06474-f011:**
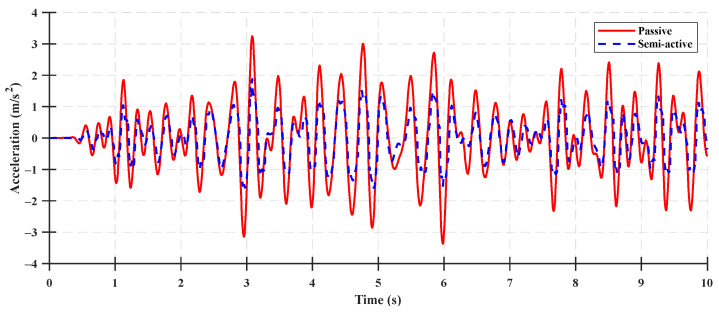
Comparison between the cabin vertical acceleration of the semi-active suspension and rubber mount using the ISO 8608 standard Class A with respect to a tractor speed of 5 km/h.

**Figure 12 sensors-23-06474-f012:**
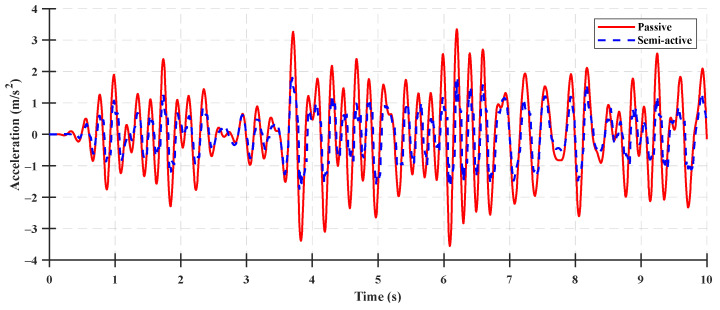
Comparison between the cabin vertical acceleration of the semi-active suspension and rubber mount using the ISO 8608 standard Class B with respect to a tractor speed of 5 km/h.

**Figure 13 sensors-23-06474-f013:**
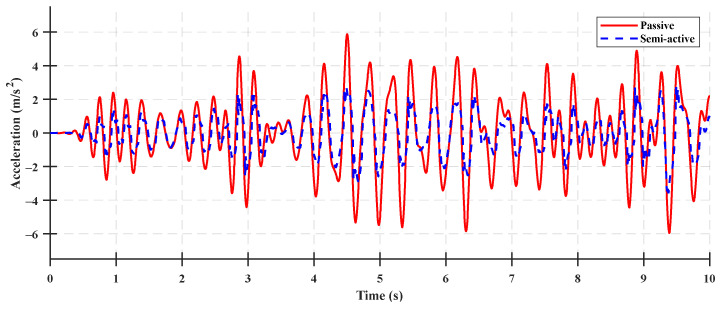
Comparison between the cabin vertical acceleration of the semi-active suspension and rubber mount using the ISO 8608 standard Class C with respect to a tractor speed of 5 km/h.

**Table 4 sensors-23-06474-t004:** Performance indices of the ISO 8608 standard road profile Class A.

Performance Index	ISO Standard Road Profile Class A		
	Passive (m/s^2^)	Semi-Active (m/s^2^)	Reduction Ratio (%)
Peak value of cabin vertical acceleration	3.37	1.89	43.74
RMS value of cabin vertical acceleration	1.16	0.68	41.12

**Table 5 sensors-23-06474-t005:** Performance indices of the ISO 8608 standard road profile Class B.

Performance Index	ISO Standard Road Profile Class B		
	Passive (m/s^2^)	Semi-Active (m/s^2^)	Reduction Ratio (%)
Peak value of cabin vertical acceleration	3.55	1.81	48.97
RMS value of cabin vertical acceleration	1.24	0.70	43.12

**Table 6 sensors-23-06474-t006:** Performance indices of the ISO 8608 standard road profile Class C.

Performance Index	ISO Standard Road Profile Class C		
	Passive (m/s^2^)	Semi-Active (m/s^2^)	Reduction Ratio (%)
Peak value of cabin vertical acceleration	5.94	3.58	39.78
RMS value of cabin vertical acceleration	2.16	1.14	47.06

The tractor with a rear rubber mount was a passive suspension system where the damping force was determined using the constant damping coefficient and the relative velocity of the cabin and body. The suspension control algorithm was not applied to the tractor with a rear rubber mount, as shown in Figure 11, Figure 12 and Figure 13. For the tractor with a rear semi-active suspension, the damping coefficient of the semi-active suspension was determined as the target damping force using the controller in the LQG control algorithm. The semi-active suspension system with an LQG controller was evaluated by comparing the performance indices of a system with and without the application of the control algorithm (under ISO 8608 standard Classes A–C). With the designed LQG controller, the peak vertical acceleration of the cabin decreased by 43.74% from 3.37 m/s^2^ to 1.89 m/s^2^, and the RMS vertical acceleration of the cabin decreased by 41.12% from 1.16 m/s^2^ to 0.68 m/s^2^, under the Class A road profile. For the Class B road profile, the peak vertical acceleration of the cabin decreased by 48.97% from 3.55 m/s^2^ to 1.81 m/s^2^, and the RMS value of the vertical acceleration of the cabin decreased by 43.12% from 1.24 m/s^2^ to 0.70 m/s^2^. For the Class C road profile, the peak vertical acceleration of the cabin decreased by 39.78% from 5.94 m/s^2^ to 3.58 m/s^2^, and the RMS vertical acceleration of the cabin decreased by 47.06% from 2.16 m/s^2^ to 1.14 m/s^2^. The LQG controller of the semi-active suspension system was acceptable for improving ride comfort, which reduced the reduction ratio of the vertical acceleration of the cabin from 39% to 48%. Although the state observer was designed using a limited number of measurement inputs, its accuracy was 79% under the ISO standard profile condition.

The performance of the designed control algorithm was examined by conducting a parametric study in Figure 14 and Figure 15. The decreases in the peak and RMS values of the cabin acceleration were evaluated with respect to different tractor working speeds in the range 3–10 km/h under the ISO standard profile condition. The decrease in peak acceleration of the cabin was in the range 35.64–57.34%, and the decrease in the RMS value of acceleration was in the range 30.98–53.95% under the tractor working conditions.

## 4. Conclusions

In this study, a semi-active suspension control algorithm was designed to improve the ride comfort in an agricultural tractor cabin. The study can be summarized as follows:

(1) To design a semi-active suspension control algorithm, a half-car suspension model of a tractor, based on the parameters of the actual tractor, was chosen as the plant model. The rear suspension deflection of the designed plant model was represented by Kalman-filter-based state observer feedback state variables. The estimated state variables were used as the state variables for the LQR control.

(2) An accuracy index was defined to validate the performance of the Kalman-filter-based state observer of the designed LQG controller. The designed state observer was validated under different road profiles (ISO 8608 standard classes) in simulations and was confirmed as acceptable to use for the state feedback of LQR control.

(3) Performance indices were defined to validate the performance of the designed the LQG controller, through which the performance of a tractor equipped with a rear rubber mount was compared with that of a tractor equipped with a rear semi-active suspension. The peak vertical acceleration of the cabin was reduced by up to 48.97%, and the RMS vertical acceleration of the cabin was reduced by up to 47.06% under the road profile in the ISO 8608 standard class.

Therefore, the performance of the LQG controller designed in this study improved the ride comfort of the tractor (under the ISO 8608 standard class). Further critical points are presented as follows.

(1) The improvement in the state observer for the performance enhancement of the LQG controller was as follows. The state observer of the LQG controller designed in this study had a limited number of measurement inputs and only measured the rear suspension deflection. The plant model and output variables in this study posed a challenge with respect to the state variable estimation of the designed state-space model as the plant model. The performance and accuracy of the state observer can be improved by installing additional sensors on the tractor, and the performance of the LQG controller can be improved using the high-accuracy estimated state variable as the feedback state of the LQR controller.

(2) The plant model for the semi-active suspension controller design software was developed as follows. The semi-active suspension system of the plant model was designed such that the suspension force was generated according to the current of the valve controller without time delays. However, the actual semi-active suspension system underwent two delays until the suspension force was generated by the current generated using the valve controller. The first was a delay that occurred in the valve controller (which controlled the proportional control valve of the semi-active suspension). The second was a delay caused by the dynamic characteristics of the proportional control valve. The delays involved in determining the suspension force are critical factors influencing the performance of semi-active suspension control. Therefore, the plant model should be improved by implementing delays similar to the characteristics of an actual semi-active suspension. Overall, this study can serve as a basis for the application of the control algorithm to systems with similar characteristics and can reduce the system cost in the process of developing a tractor suspension control algorithm.

## Figures and Tables

**Figure 1 sensors-23-06474-f001:**
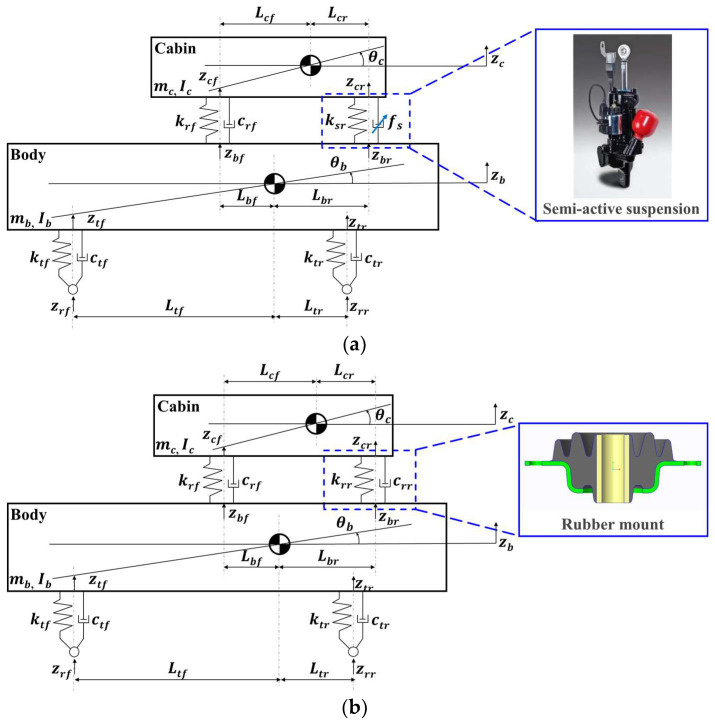
Half-car suspension model for tractors used in this study: (**a**) rear semi-active suspension model and (**b**) rear rubber mount model.

**Figure 2 sensors-23-06474-f002:**
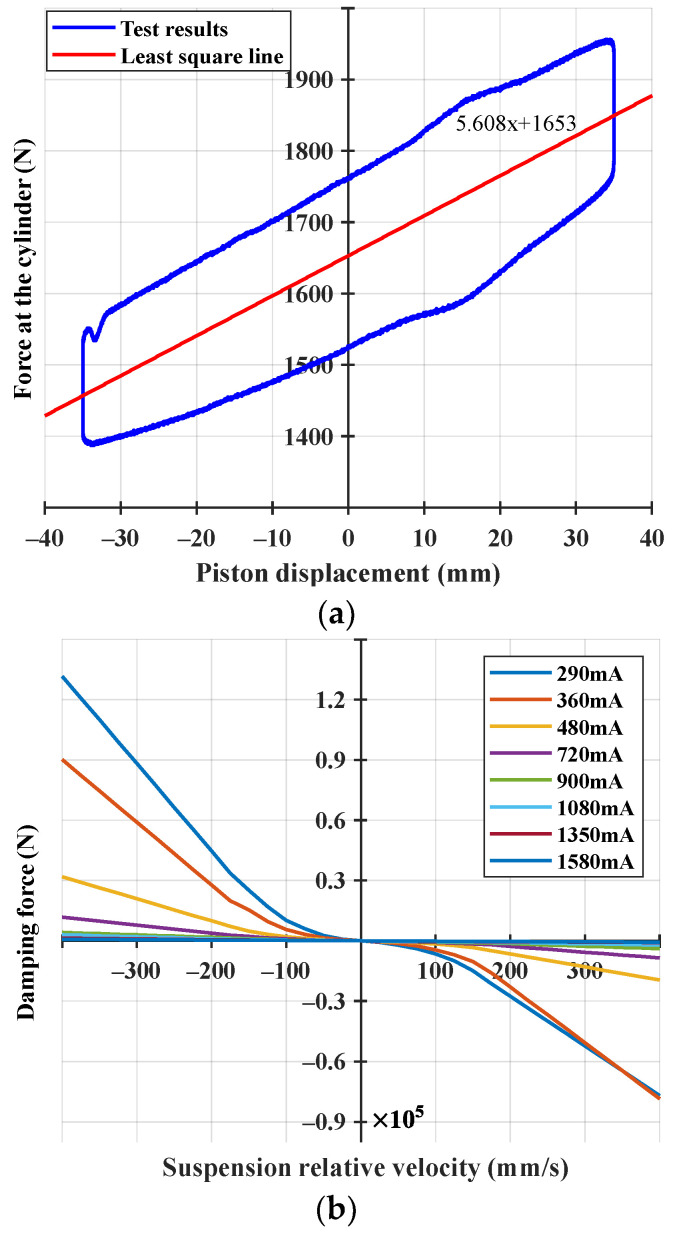
Semi-active suspension characteristics: (**a**) stiffness characteristics and (**b**) damping characteristics with respect to the applied valve current.

**Figure 3 sensors-23-06474-f003:**
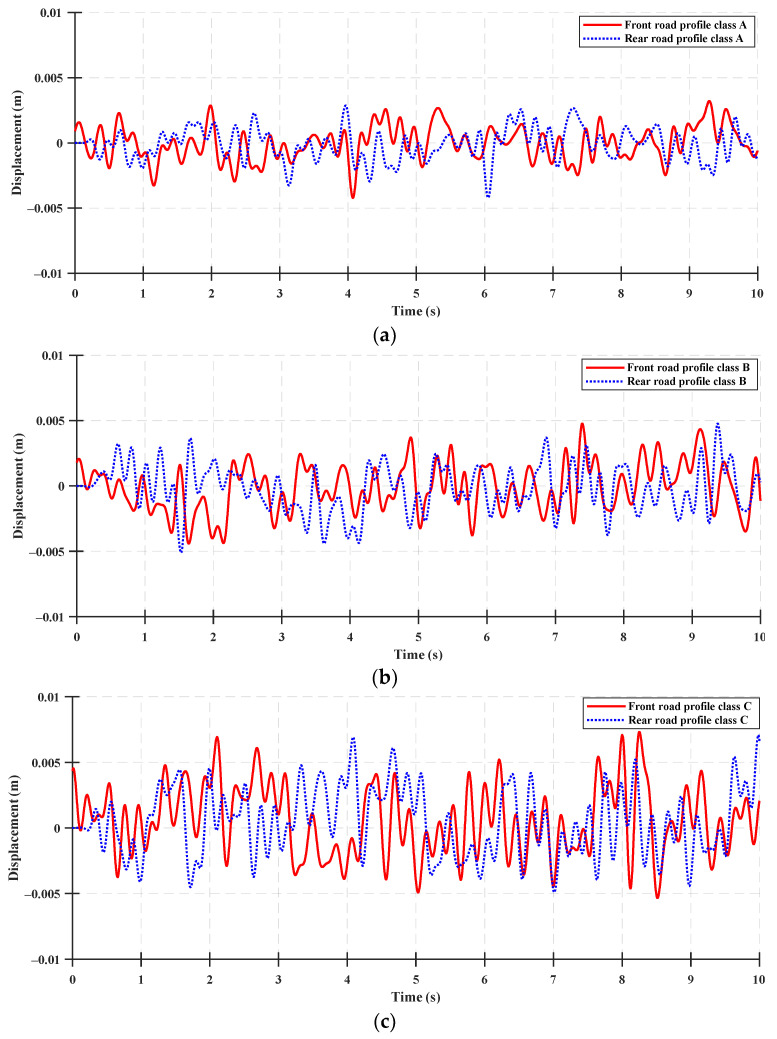
ISO 8608 standard road profile with respect to a tractor speed of 5 km/h: (**a**) Class A, (**b**) Class B, and (**c**) Class C.

**Figure 4 sensors-23-06474-f004:**
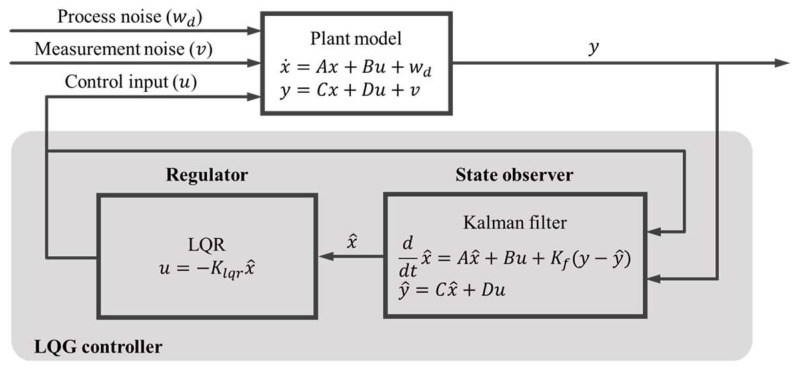
Schematic of an LQG controller.

**Figure 5 sensors-23-06474-f005:**
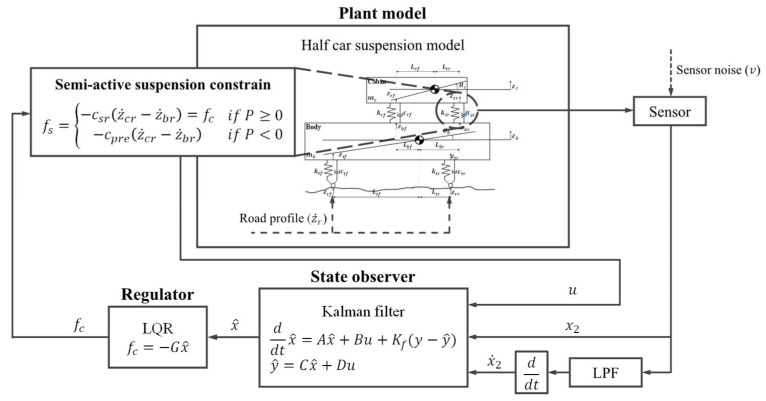
Flowchart of the LQG controller for the tractor semi-active suspension system.

**Figure 6 sensors-23-06474-f006:**
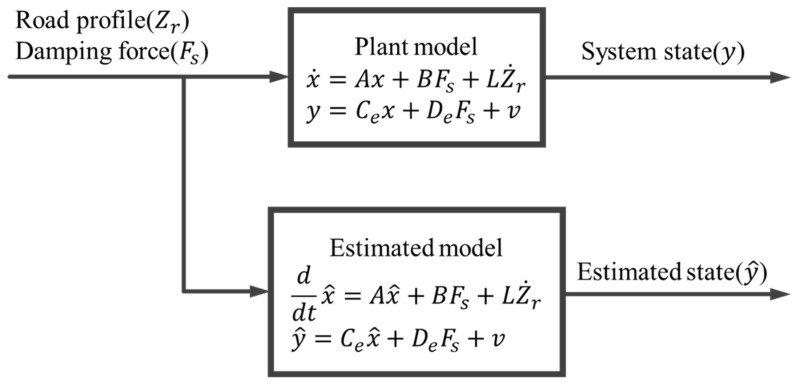
Schematic of the state observer validation.

**Figure 7 sensors-23-06474-f007:**
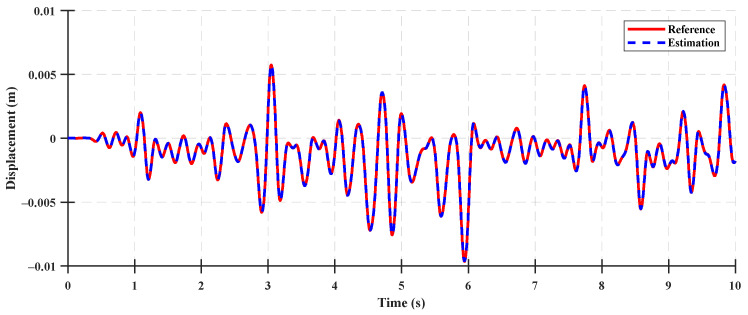
Comparison between the state estimation results for the rear suspension deflection of the system and the estimated states on a Class A road.

**Figure 8 sensors-23-06474-f008:**
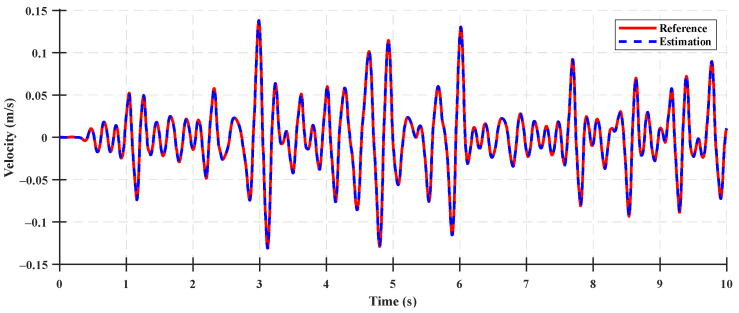
Comparison between the state estimation results for the rear suspension relative velocity of the system and the estimated states on a Class A road.

**Figure 9 sensors-23-06474-f009:**
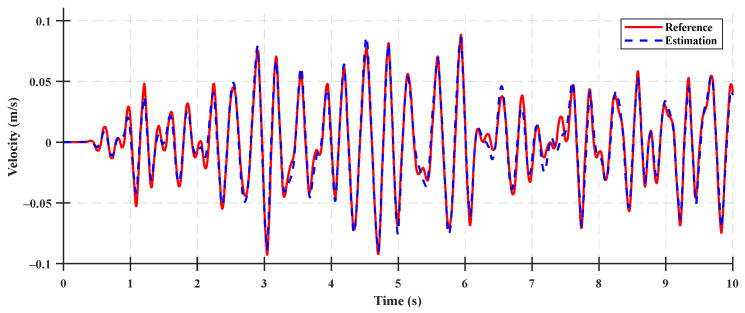
Comparison between the state estimation results for the cabin vertical velocity of the system and the estimated states on a Class A road.

**Figure 10 sensors-23-06474-f010:**
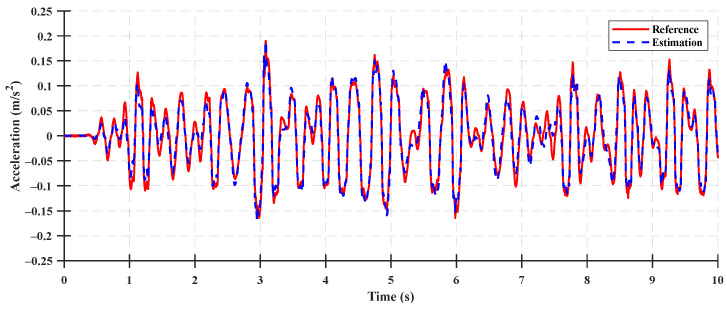
Comparison between the state estimation results for the cabin vertical acceleration of the system and the estimated states on a Class A road.

**Figure 14 sensors-23-06474-f014:**
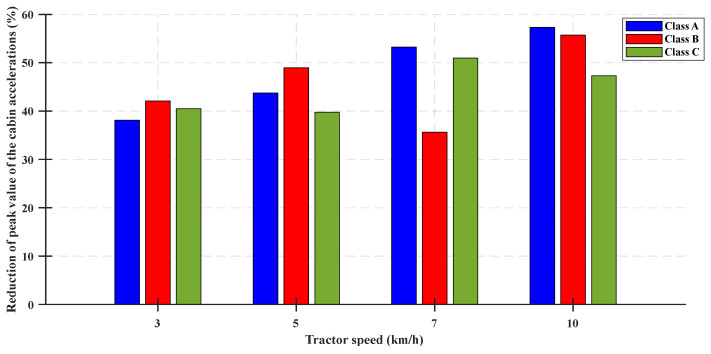
Reduction of peak value of the cabin vertical acceleration considering the ISO 8608 standard for the tractor speed.

**Figure 15 sensors-23-06474-f015:**
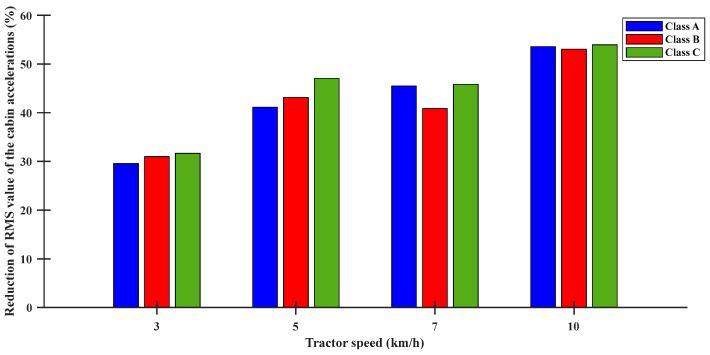
Reduction of RMS value of the cabin vertical acceleration considering the ISO 8608 standard for the tractor speed.

**Table 1 sensors-23-06474-t001:** Road profiles classified using the ISO standard [19].

Road Class	Degree of Roughness $G_{d}\left(n_{0}\right)\;\left(10^{-6}m^{3}\right)$, Where $n_{0}\;=\;0.1\;\mathrm{cycle}/m$		
	Lower Limit	Geometric Mean	Upper Limit
A	-	16	32
B	32	64	128
C	128	256	512

**Table 2 sensors-23-06474-t002:** Tractor parameters for simulation [18].

Parameter		Symbol	Value	Unit
Cabin	Mass	$m_{c}$	275.0	kg
	Inertia	$I_{c}$	91.4	${\mathrm{kgm}}^{2}$
Body	Mass	$m_{b}$	2526.5	kg
	Inertia	$I_{b}$	1679.0	${\mathrm{kgm}}^{2}$
Front tire	Stiffness	$k_{tf}$	570,690	N/m
	Damping	$c_{tf}$	4394	Ns/m
	length	$L_{tf}$	1.508	m
Rear tire	Stiffness	$k_{tr}$	483,790	N/m
	Damping	$c_{tr}$	2951	Ns/m
	length	$L_{tr}$	1.244	m
Front rubber mount	Stiffness	$k_{sf}$	1,132,550	N/m
	Damping	$c_{sf}$	2208	Ns/m
	length	$L_{cf}$	0.799	m
		$L_{bf}$	0.041	m
Rear rubber mount	Stiffness	$k_{rr}$	618,599	N/m
	Damping	$c_{rr}$	1278	Ns/m
	length	$L_{cr}$	0.667	m
		$L_{br}$	1.425	m
Rear semi-active suspension	Stiffness	$k_{sr}$	5608	N/m

**Table 3 sensors-23-06474-t003:** Accuracy index results for state estimation on a Class A road.

System State	Accuracy Index of Estimated State in a Class A Road (%)
Rear suspension deflection	99.26
Relative velocity of rear suspension	99.54
Cabin vertical velocity	83.31
Cabin vertical acceleration	79.87

## Data Availability

We are not able to make our data available to the public because of privacy constraints.

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
