# Peer review of "Optimal Control of Semi-Active Suspension for Agricultural Tractors Using Linear Quadratic Gaussian Control"

_sensors, 2023, doi:10.3390/s23146474_

Round 1

Reviewer 1 Report (Previous Reviewer 3)

No further comments.

The paper is understandable and well-organized.

Author Response

We would like to thank the reviewer for taking the time to review our manuscript.

Reviewer 2 Report (Previous Reviewer 2)

The paper is well revised, I have no further comments.

Author Response

We would like to thank the reviewer for taking the time to review our manuscript.

Reviewer 3 Report (Previous Reviewer 1)

I read the paper, and even though it has been improved compared with the previous version, however, there are some issues to develop.

1/ Why eq 35 is introduced in the paper?

2/ In matrix C after eq 38, shows that only the second component is measurable. Is the case in practice for the system?

3/ Eq 43, represents the steady state of the Ticatti equation. How to justify the convergence of Y?

4/ In the simulation, the authors have to present step-by-step the design procedure and the found values of different variables.

5/ The variable notations should be indicated in the figures.

6/ What happens if you consider an initial condition different to zero?

7/ Kalman filter assumes white noise inputs. How to verify that \dot z_r can be considered as white noise?  Is there a correlation between the sensor noise and \dot z_r ?

May be improved.

Author Response

Thank you for providing these insights. Response to the reviewer have been prepared and are attatched in  a separate file.

Reviewer 4 Report (New Reviewer)

I would like to commend the authors on their good job of conducting the research and presenting findings. The paper is well-written and could be published with minor editorial changes. Please kindly check the language for small grammatical and expression errors. Delete all yellow highlights before publishing.

minor editing needed

Author Response

We would like to thank the reviewer for taking the time to review our manuscript. We have removed the yellow highlights in the Manuscript. This manuscript has been proofread and edited by Editage, a professional English editing service. The editing certificate for the paper was an attachment file.

Round 2

Reviewer 3 Report (Previous Reviewer 1)

The authors responded to my comments

No comments

This manuscript is a resubmission of an earlier submission. The following is a list of the peer review reports and author responses from that submission.

Round 1

Reviewer 1 Report

Comments to the Author

The paper investigates the optimal control of semi-active suspension for agricultural tractors using linear quadratic Gaussian control. Please find below some suggestions for further improvement.

1.      A significant language problem in the paper needs to be addressed, especially the use of the past.

2.      This sentence is too long and loses its meaning as a result. For example, the meaning of this sentence is not clear “Numerous studies related to the development of suspension-control algorithms for vehicles have been conducted because semi-active or active suspension systems were introduced before tractors.”

3.      The motivation of this study has to be clearly highlighted in the introduction.

4.      There is no difference between the two figures on page 4!

5. All parameters in the model must be defined. (csr in eq 7 is not specified)

6.      Eq 11 is not compatible with eq 7.

7.      X4 is not compatible with \dot x2 in 23.

8.      What is E and x_k in 41

9.      The dimensions of the matrices in Eq 46 are not compatible.

10. In conclusion, the paper presents a classic method for a linear system. It should be deeply revised where the design steps should be included in an algorithm and the different gains of the filter and controller should be added. The control law should be depicted.

Comments to the Author

The paper investigates the optimal control of semi-active suspension for agricultural tractors using linear quadratic Gaussian control. Please find below some suggestions for further improvement.

1.      A significant language problem in the paper needs to be addressed, especially the use of the past.

2.      This sentence is too long and loses its meaning as a result. For example, the meaning of this sentence is not clear “Numerous studies related to the development of suspension-control algorithms for vehicles have been conducted because semi-active or active suspension systems were introduced before tractors.”

3.      The motivation of this study has to be clearly highlighted in the introduction.

4.      There is no difference between the two figures on page 4!

5. All parameters in the model must be defined. (csr in eq 7 is not specified)

6.      Eq 11 is not compatible with eq 7.

7.      X4 is not compatible with \dot x2 in 23.

8.      What is E and x_k in 41

9.      The dimensions of the matrices in Eq 46 are not compatible.

10. In conclusion, the paper presents a classic method for a linear system. It should be deeply revised where the design steps should be included in an algorithm and the different gains of the filter and controller should be added. The control law should be depicted.

Reviewer 2 Report

The paper is concerned with the optimal control of semi-active suspension for agricultural trac-2 tors using linear quadratic Gaussian control. After reading the whole paper, I have the following comments:

1. The presentation of this paper is a mess, those equations should be well improved as many of them exceed the page size.

2. The main results contain errors, see, e.g., the observer in Figure 4 is not Kalman filter, it is a traditional Luenberger observer.

3. There is any results on the convergence analysis of the estimation error system.

4. Reference format is not unified.

The presentation of this paper is a mess, those equations should be well improved as many of them exceed the page size.

Reviewer 3 Report

1. There are some strong concerns about the contribution of the study, would you like to state explicitly as which aspect of controller did you explored?

2. The work seems like the utilization of the existing well-established literature on LQR controller and observers. The work can be classified as a textbook stuff.

3. I believe that the authors are claiming that they worked upon a new type of tractor suspension model. How would you justify that, the MSD systems in different configurations have been around for many decades?

4. Figure 1 a and 1 b seems to be identical and they are not readable as well.

The paper is written in a clear and understandable manner.
